# Transmit–Receive Sparse Synthesis of Linear Frequency Diverse Array in Range-Angle Space Using Genetic Algorithm

**DOI:** 10.3390/s23063107

**Published:** 2023-03-14

**Authors:** Yanhong Xu, Xiao Huang, Anyi Wang

**Affiliations:** Xi’an Key Laboratory of Network Convergence Communication, College of Communication and Information Engineering, Xi’an University of Science and Technology, Xi’an 710054, China

**Keywords:** beampattern synthesis, transmit–receive mode, frequency diverse array (FDA), sparse array, range and angle domains, genetic algorithm (GA)

## Abstract

Unlike conventional phased array (PA), frequency diversity array (FDA) can perform the beampattern synthesis not only in an angle dimension but also in a range dimension by introducing an additional frequency offset (FO) across the array aperture, thus greatly enhancing the beamforming flexibility of an array antenna. Nevertheless, an FDA with uniform inter-element spacing that consists of a huge number of elements is required when a high resolution is needed, which results in a high cost. To substantially reduce the cost while almost maintaining the antenna resolution, it is important to conduct a sparse synthesis of FDA. Under these circumstances, this paper investigated the transmit–receive beamforming of a sparse-fda in range and angle dimensions. In particular, the joint transmit–receive signal formula was first derived and analyzed to resolve the inherent time-varying characteristics of FDA based on a cost-effective signal processing diagram. In the sequel, the GA-based low sidelobe level (SLL) transmit–receive beamforming of the sparse-fda was proposed to generate a focused main lobe in a range-angle space, where the array element positions were incorporated into the optimization problem. Numerical results showed that 50% of the elements can be saved for the two linear FDAs with sinusoidally and logarithmically varying frequency offsets, respectively termed as sin-FO linear-FDA and log-FO linear-FDA, with only a less than 1 dB increment in SLL. The resultant SLLs are below −9.6 dB, and −12.9 dB for these two linear FDAs, respectively.

## 1. Introduction

First proposed by Antonik in 2006 [1], frequency diverse array (FDA) has been receiving extensive attention due to its unique range-angle-dependent beampattern. Compared with conventional phased array (PA), FDA possesses an additional degrees-of-freedom (DOFs) in range dimension, thereby substantially enhancing the flexibility of array beamforming [2,3,4]. Specifically, FDA can provide potential superiority in precise target location [5,6], and range-dependent anti-jamming [7,8] over conventional PA which can only perform the beamforming in an angle dimension [9]. At present, enormous investigations have been conducted on the potential applications of FDA [10,11,12,13,14,15,16,17,18,19,20,21]. In particular, the concept of FDA was introduced into the millimeter wave imaging systems to reduce the sampling rates in [10]. In [11], a high-speed user-centric beampattern synthesis approach was proposed via FDA to maintain communication quality. The benefits of FDA in forward-looking radar GMTI were analyzed in [12]. FDA was first introduced from radar to lamb wave in [13], which opens up the possibility of realizing range-angle focusing in damage localization. In [14], a security-enhanced, spectral-efficient, and power-efficient multi-beam wireless communication scheme was proposed based on random FDA. In [15], FDA was utilized in ambient backscatter communication to enhance the channel capacity and detection performance. A mixed near-field and far-field target localization method was presented in [16], which can obtain both direction-of-arrival (DOA) and range information of the targets by utilizing symmetric nested FDA. A dual-mode array radar scheme, namely PA and FDA cooperated radar, was developed in [17] to provide unambiguous range and angle estimation, which is of significant importance in wide region surveillance. Additionally, it was pointed out in [18,19,20,21] that FDA would have greater advantages over conventional PA in many future applications, such as multitask radar, satellite, navigation systems, etc.

Apart from the above investigations, many researchers have devoted themselves to the analysis and design of the transmit beampattern of linear-FDAs or planar-FDAs [22,23,24,25,26,27,28,29,30,31,32]. In the simplest form of FDA, a linearly varying frequency offset (lin-FO) is applied across the aperture of a uniform linear array (ULA), resulting in an S-shaped pattern with multiple maximum values in range-angle space which is not suitable for accurate target positioning. Obviously, the FO has great influence on the beampattern of FDA. Therefore, using different types of FO can achieve the purpose of a range-angle information decoupled pattern. In [22,23], non-uniform FOs were utilized to achieve a single maximum in a range-angle space. A symmetric FDA with non-uniform FO and inter-element spacing was proposed in [24] to obtain a dot-shaped beampattern with better performance. An FDA with modified sinusoidal FO (MSin-FDA) was proposed in [25] to obtain a narrow pencil beampattern. By adjusting the FO of MSin-FDA, a better performance was achieved in spatial focusing, sidelobe suppression, and resolution. In [26], an FDA framework based on Taylor window FO was proposed. An FDA transmitter structure with random logarithmically increasing FO (log-FDA) was proposed in [27] to achieve a lower sidelobe level (SLL) and higher detection resolution simultaneously. Nonlinear FOs were utilized in conformal FDA to achieve focused pattern in range-angle space [28,29]. An FDA transmit beampattern design scheme was proposed in [30] with both a random permutated and power-increasing FO. In [31], the FO was optimized by a genetic algorithm (GA) to synthesize a dot-shaped transmit beampattern with better target positioning performance. In [32], a bat algorithm (BA)-based synthesis technique was proposed for uncouple range-angle beamforming by optimizing the FO and the current excitations. A range-angle transmit beampattern synthesis method for FDA based on FO optimization was proposed in [33], where particle swarm optimization (PSO) was adopted in the FDA element frequency increment design. Although the range-angle-decoupled beampattern can be obtained for FDA, the inherent time-varying issue still exists in most of the above investigations, which increases the difficulty of the mainbeam direction control. To alleviate this problem, an equivalent transmit beamforming scheme was proposed in [34] to achieve a range-angle decoupled beampattern with low SLL, where the time-varying terms are handled at the receiver. An enhanced transmit–receive beamforming method was proposed in [35] with predesigned FOs to achieve time-invariant and symmetric beampatterns with only a single maximum in the range-angle space. In [36], an FDA radar transceiver system was proposed for multi-target localization where the subarray-based FDA and full-band FDA were chosen as the transmitter and receiver, respectively. In [37], an FDA antenna system for target range-angle imaging was proposed, where the high-resolution imaging of the target in range-angle space was achieved by the cooperative transmit–receive beamforming. From the above descriptions, it can be concluded that the joint transmit–receive beamforming can effectively deal with the inherent time-varying problem of FDA when the system parameters are properly designed.

It is known that array beamforming is a powerful technique for enhancing the performance of array antennas. For an FDA, the array geometry also plays an important role in array beamforming apart from the FO. The sparse synthesis of array antenna can substantially reduce the cost while almost maintaining the antenna resolution, which is of great significance in practical scenarios [38]. At present, most research is conducted on the sparse synthesis of PA, while investigations on the sparse synthesis of FDA are quite limited. In [39], a decoupled virtual two-dimensional sparse-fda beampattern was synthesized by jointly optimizing the frequency and array position where 40% of the elements were saved. An algorithm based on group sparse recovery and convex optimization was proposed for sparse multicarrier FDA in [40], where 25% of the elements were saved. The artificial bee colony (ABC) algorithm was adopted in [41] for sparse-fda to achieve a spatial-focused beampattern. An efficient beampattern synthesis approach was proposed in [42] via the matrix pencil method. Basically, in the transmit mode, the FDA beampattern is range-angle-time dependent, and the time-varying issue always exists. Existing research studies are mainly focused on the transmit beampattern of sparse-fda and the time-varying problem is not handled. Aimed at achieving a time-invariant range-angle information decoupled pattern while greatly reducing the cost and maintaining the antenna resolution simultaneously, this paper proposed a transmit–receive range-angle two-dimensional beamforming approach of sparse-FDA based on GA which is one of the effective global optimization algorithms and has been widely utilized to optimize array architecture at a low complexity [43,44,45]. Based on the transmit–receive ideology presented in [29] and inspired by the effect of nonlinear FO in range-angle-decoupled beamforming of FDA, such as sin-FO in [25] and log-FO in [35], a transmit–receive sparse synthesis approach for linear-FDA was presented in this article. In particular, the joint transmit–receive signal formula was first derived and analyzed to resolve the inherent time-varying characteristics of FDA based on the cost-effective signal processing diagram. The theoretical model was then constructed to realize the transmit–receive sparse synthesis of the two linear-FDAs with sin-FO and log-FO based on GA with focused mainbeam in the desired range-angle space and low SLL in the region out-of-interest simultaneously.

The rest of this paper is arranged as follows. In Section 2, the joint transmit–receive signal formula was derived and analyzed to resolve the inherent time-varying characteristics of FDA. In Section 3, the low-SLL transmit–receive pattern sparse synthesis of linear-FDA with non-linearly varying (increasing or decreasing) FO (nonlin-FO) was constructed based on GA. Numerical results are provided in Section 4. Finally, Section 5 concludes the work of this paper.

## 2. Transmit–Receive Signal Model for Linear-FDA with Nonlin-FO

In this section, the uniformly spaced linear FDA, termed as linear-FDA, with nonlin-FO is chosen as an illustration. In specific, its structure is described in Section 2.1 and the corresponding transmit–receive signal model is derived and analyzed in Section 2.2.

### 2.1. Structure of Linear-FDA with Nonlin-FO

As mentioned, a small FO exists between the adjacent elements of FDA. As depicted in Figure 1, a linear-FDA composed of N elements with inter-element spacing d is taken into consideration.

Taking the first element as reference, the operational frequency of the signal transmitted from the *n*th element has the form of
(1)$$f_{n}=f_{0}+\Delta f_{n},n=1,\ldots ,N$$
where $f_{0}$ is the reference frequency and $\Delta f_{n}$ is the FO applied in the *n*th element. For lin-FO, $\Delta f_{n}=\left(n-1\right)\Delta f$ with Δf denoting a small constant, which is negligible compared with $f_{0}$. As aforementioned, nonlin-FO should be adopted for linear-FDA to achieve the range-angle information decoupled beampattern. Inspired by the sin-FO in [25] and log-FO in [35], two linear-FDAs, termed as sin-FO linear-FDA and log-FO linear-FDA, are adopted as an illustration. In particular, the FOs of the *n*th element in these two scenarios can be separately expressed as
(2a)$$\Delta f_{n}=-9\Delta f\sin\left(\left(n\right)/38.8\right),n=1,2,\ldots ,N$$
and
(2b)$$\Delta f_{n}=-\Delta f{\left[\ln\left(n\right)\right]}^{1.5},n=1,2,\ldots ,N$$
where sin(*n*) and ln(*n*) denote the sinusoidal and natural logarithm function of *n*, respectively. Note that Equation (2a) is set in this form to avoid the appearance of two signals of identical frequency. To clearly show the variation trends of the abovementioned three kinds of FO, Figure 2 provides their corresponding curves with respect to the element number.

Therefore, the signal transmitted from the *n*th element of linear-FDA can be expressed as follows.
(3)$$p_{n}\left(t\right)=\exp\left(j2\pi f_{n}\left(t-\frac{R_{n}}{c}\right)\right)$$
where c is the speed of light, $R_{n}$ denotes the distance from the *n*th element to the target and can be approximately expressed as
(4)$$R_{n}\approx R-\left(n-1\right)d\cos\left(\theta \right)$$
where R and θ, respectively, denote the range and angle parameters of the target in the far-field region of the linear-FDA. Herein, it should be highlighted that to avoid grating lobes, $d=\lambda_{\min}/2$ where $\lambda_{\min}=c/f_{0}$ since a set of negative FOs is used to ensure that $f_{0}$ is always the maximum frequency according to Equation (2).

### 2.2. Transmit–Receive Signal Model of Linear-FDA with Nonlin-FO

In this subsection, two modes, i.e., the transmit and receive modes, are jointly considered to break the inherent time-varying characteristic of FDA. According to the above formula in Section 2.1, the signal transmitted by the *n*th transmit element, reflected by the far-field target in (R,θ), and received by the *m*th receive element can be written as
(5)$$p_{m,n}\left(t,R,\theta \right)=\exp\left\{j\left[2\pi f_{n}\left(t-\frac{R_{n}}{c}-\frac{R_{m}}{c}\right)\right]\right\}$$
where $R_{m}\approx R-\left(m-1\right)d\cos\left(\theta \right),m=1,2,\ldots ,N$.

Therefore, the output of the *m*th receive element is the summation of the *N* transmitted signals, i.e.,
(6)$$q_{m}\left(t,R,\theta \right)=\sum_{n=1}^{N}p_{m,n}\left(t,R,\theta \right).$$

Note that the time-varying terms still exist, and the signals are not separated either. Figure 3 depicts a cost-effective diagram of the receive signal processing chain of the *m*th receive element, which is merely mentioned in [29] as Figure 2b. In the following, the specific signal model is derived and analyzed based on this diagram.

As depicted in this figure, it is seen that the received signal of the *m*th receive element is firstly amplified by a low-noise amplifier (LNA), and then mixed with a mixer with local frequency of $f_{0}$. Therefore, the output signal of the mixer can be expressed as
(7)$$\begin{array}{cc} {\hat{q}}_{m}\left(t,R,\theta \right) & =q_{m}\left(t,R,\theta \right)\times \exp\left(-j2\pi f_{0}t\right)\\ \; & =\sum_{n=1}^{N}\exp\left\{j2\pi \left[\Delta f_{n}t-\frac{f_{n}\left(R_{n}+R_{m}\right)}{c}\right]\right\} \end{array}$$

After this implementation, the signal is sampled by an ADC. Then a series of digital mixers (DMs) with bandwidth of $\Delta f_{i},i=1,\ldots ,N$ are utilized to compensate the time-varying terms due to FO, i.e., $\exp\left\{j2\pi \Delta f_{n}t\right\},n=1,\ldots ,N$. Thereby, the output signal of the *i*th digital mixer has the form of
(8)$$\begin{array}{cc} {\tilde{q}}_{m,i}\left(t,R,\theta \right) & ={\hat{q}}_{m}\left(t,R,\theta \right)\times \exp\left(-j2\pi \Delta f_{i}t\right)\\ \; & =\sum_{n=1}^{N}\exp\left\{j2\pi \left[\left(\Delta f_{n}-\Delta f_{i}\right)t-\frac{f_{n}\left(R_{n}+R_{m}\right)}{c}\right]\right\}\\ \; & =\exp\left\{\frac{-j2\pi f_{i}\left(R_{i}+R_{m}\right)}{c}\right\}+\sum_{n=1,n\neq i}^{N}\exp\left\{j2\pi \left[\left(\Delta f_{n}-\Delta f_{i}\right)t-\frac{f_{n}\left(R_{n}+R_{m}\right)}{c}\right]\right\} \end{array}$$

From Equation (8), it is seen that the time-varying terms have not been removed. In the sequel, the signals are filtered by a series of identical low-pass filters (LPFs). Then the output signal of the *i*th LPF can be formulated as
(9)$$\begin{array}{cc} {\bar{q}}_{m,i}\left(R,\theta \right) & ={\tilde{q}}_{m,i}\left(t,R,\theta \right)*h\left(t\right)\\ \; & =\exp\left\{\frac{-j2\pi f_{i}\left(R_{i}+R_{m}\right)}{c}\right\}\\ \; & =\exp\left\{\frac{-j2\pi f_{i}\left[2R-\left(m+i-2\right)d\cos\left(\theta \right)\right]}{c}\right\} \end{array}$$
where h(t) represents the temporary response of the LPF and the ∗ denotes the convolution operator. From Equation (9), it is seen that the *N* transmitted signals received by the *m*th receive element are separately from each other. According to Equations (8) and (9), it is seen that the time-varying terms are eliminated with a series of DMs and LPFs. Compared with the diagram presented in [34,35], the hardware complexity in this diagram is significantly reduced since only one mixer and one ADC are utilized. As the ADC is usually the most expensive component, this diagram can greatly reduce the cost.

Consequently, the output signal of the *m*th receive processing chain can be constructed as
(10)$${\overset{⌣}{q}}_{m}\left(R,\theta \right)=\sum_{i=1}^{N}w_{m,i}{\bar{q}}_{m,i}\left(R,\theta \right)$$
after summing up the weighted signals from each channel where $w_{m,i}$ denotes the weight of the *i*th channel in the *m*th chain.

Therefore, the resultant transmit–receive array factor of the linear-FDA can be written as
(11)$$\mathrm{AF}\left(R,\theta \right)=\sum_{m=1}^{N}\upsilon_{m}{\overset{⌣}{q}}_{m}\left(R,\theta \right)=\sum_{m=1}^{N}\sum_{i=1}^{N}b_{m,i}{\bar{q}}_{m,i}\left(R,\theta \right)$$
where $b_{m,i}=v_{m}w_{m,i}$ with $v_{m}$ representing the weight of the *m*th processing chain which will be incorporated into the optimization problem in Section 3 for the transmit–receive beamforming of sparse-fda.

## 3. Transmit–Receive Beamforming of Sparse-FDA Based on GA

Inspired by the fact that GA can optimize array architecture at a low complexity [43,44,45], it is adopted for the transmit–receive beamforming of sparse-FDA in this section. In particular, the transmit–receive power pattern of linear-FDA is formulated in Section 3.1 with the transmit–receive weight vector and steering vector reconstructed to facilitate the establishment of the optimization model, which is presented in the following subsection, i.e., Section 3.2. Note that the resultant sparse-FDA with desired properties is obtained on the basis of the linear-FDA.

### 3.1. Transmit–Receive Power Pattern of Linear-FDA

Represent the transmit–receive weight vector and the steering vector as
(12)$$b={\left[b_{1}^{T},\ldots ,b_{m}^{T},\ldots ,b_{N}^{T}\right]}^{T}$$
and
(13)q(R,θ)=[q1T(R,θ),…,qmT(R,θ),…,qNT(R,θ)]T
respectively, where bm=[bm,1,…,bm,n,…,bm,N]H∈ℂN×1,m=1,…,N, qm(R,θ)=[q¯m,1(R,θ),…,q¯m,n(R,θ),…,q¯m,N(R,θ)]∈ℂN×1, m=1,…,N with the superscripts T and H denoting the transpose operator and conjugate transpose operator, respectively. Therefore, the transmit–receive array factor in Equation (11) can be reconstructed as
(14)AF(R,θ)=bHq(R,θ).

Consequently, the transmit–receive power pattern of linear-FDA can be expressed as
(15)P(R,θ)=|AF(R,θ)|2=bHQ(R,θ)b
where Q(R,θ)=q(R,θ)qH(R,θ)∈ℂN2×N2 denotes the transmit–receive covariance matrix of linear-FDA.

### 3.2. Construction of GA-Based Formula in Synthesis of Sparse-FDA with Focused Mainbeam and Low SLL Simultaneously

The goal is to find the optimal transmit–receive weight vector b which can makes the transmit–receive pattern P(R,θ) best approximate the desired transmit–receive pattern $P_{d}\left(R,\theta \right)$ with focused mainbeam in the desired range-angle space and low SLL in the region out-of-interest simultaneously. The best configuration of linear-FDA is determined after applying the sparse implementations under the constraint of given sparse ratio which is defined as
(16)$$\beta =\frac{N_{s}}{N}\times 100\%$$
where $N_{s}$ and N represent the element number of linear-FDA before and after applying the sparse implementation.

From the above descriptions, the transmit–receive weight vector b in Equation (12) plays a significant role in the synthesis of sparse-FDA. According to Equation (11), the entry $b_{m,n}$ of $b_{m}$ has the form of
(17)$$b_{m,n}=v_{m}w_{m,n}$$
where $v_{m}$ can be utilized to denote the working state of the *m*th receive element. The *m*th receive element is working when $v_{m}=1$, while the *m*th element is not working when $v_{m}=0$. Since the transmit array also acts as the receive array, two constraints should be satisfied during the sparse synthesis procedure, i.e., $w_{m,n}|{}_{n=m}=0$ when $v_{m}=0$ and $\mathrm{abs}\left(w_{m,n}|{}_{n=m}\right)=1$ when $v_{m}=1$ with abs(⋅) denoting the absolute value operator. After further analysis, it is seen that the first constraint can be released according to the fact that $b_{m,n}|{}_{n=m}$ is always 0 as long as $v_{m}=0$ whether $w_{m,n}|{}_{n=m}$ is equal to 0 or not. Therefore, Equation (17) can be represented as
(18)$$b_{m,n}=v_{m}\exp\left\{j\varphi_{m,n}\right\}$$
where $\varphi_{m,n}$ denote the phase of the signal transmitted by the *n*th transmit element and received by the *m*th received element. According to Equations (9), (11), (12) and (18), $\varphi_{m,n}=\exp\left\{j2\pi f_{n}/c\left[2R_{0}-\left(m+n-2\right)d\cos\left(\theta_{0}\right)\right]\right\}$ should be satisfied when the desired mainbeam is steered to $\left(R_{0},\theta_{0}\right)$. In the following, the transmit–receive weight vector b is incorporated into the optimization problem.

The weighted error function is defined as the integral of the error between the current pattern and the desired pattern in the whole observation region, which is mathematically formulated as
(19)$$F\left(b\right)=\iint_{\left(R,\theta \right)}W\left(R,\theta \right)\left|P\left(R,\theta \right)-P_{d}\left(R,\theta \right)\right|\sin\left(\theta \right)dRd\theta$$
where W(R,θ) is the weight function which is designed according to the importance over the whole observation region, and the term sin(θ) is generated due to the fact that the integration is conducted in angle dimension. Figure 4 shows the area division of the whole observation region where $\Omega =\left\{\left(R,\theta \right)|R_{\min}\leq R\leq R_{\max},0^{\circ }\leq \theta \leq 180^{\circ }\right\}$ denotes the total observation region and $\Omega_{0}=\left\{\left(R,\theta \right)|R_{a}\leq R\leq R_{b},\theta_{a}\leq \theta \leq \theta_{b}\right\}$ represents the desired mainbeam location in range-angle space. The mainbeam width in range and angle dimensions can be expressed as $R_{\mathrm{width}}=R_{b}-R_{a}$ and $\theta_{\mathrm{width}}=\theta_{b}-\theta_{a}$, respectively.

In the sequel, the objective function is established as
(20)minF(b)
the value of which would decrease as b is updated iteratively. The constraints imposed on the desired pattern can be formulated as follows.
(21)$$P_{d}\left(R,\theta \right)|{}_{\left(R,\theta \right)=\left(R_{0},\theta_{0}\right)}=1$$
(22)$$P_{d}\left(R,\theta \right)|{}_{\left(R,\theta \right)=\left(R_{a},\theta_{0}\right)\;\mathrm{and}\;\left(R,\theta \right)=\left(R_{b},\theta_{0}\right)}=0$$
(23)$$P_{d}\left(R,\theta \right)|{}_{\left(R,\theta \right)=\left(R_{0},\theta_{a}\right)\;\mathrm{and}\;\left(R,\theta \right)=\left(R_{0},\theta_{b}\right)}=0$$
(24)$$P_{d}\left(R,\theta \right)|{}_{\left(R,\theta \right)\in \Omega \;\mathrm{and}\;\left(R,\theta \right)\notin \Omega_{0}}\leq \eta$$

Equation (21) is utilized to guarantee that the generated pattern can form a maximum value at $\left(R_{0},\theta_{0}\right)$, i.e., its mainlobe should be steering to this direction; Equations (22) and (23) are the constraints imposed on the mainbeam width in range and angle dimensions, respectively; Equation (24) provides the constraint on the SLL, where η denotes the acceptable SLL.

To solve the optimization problem described in Equations (20)–(24), an effective approximation approach is adopted. In particular, the whole observation region is uniformly divided by a dense set of discrete points $\left(R_{p},\theta_{q}\right),\;p=1,\cdots ,P;\;q=1,\cdots ,Q$. Therefore, the objective function in Equation (20) can be equivalently represented as
(25)$$\min\sum_{p=1}^{P}\sum_{q=1}^{Q}\Delta_{\theta }\Delta_{R}W\left(R_{p},\theta_{q}\right)\left|P\left(R_{p},\theta_{q}\right)-P_{d}\left(R_{p},\theta_{q}\right)\right|\sin\left(\theta_{q}\right)$$
where $\Delta_{\theta }$ and $\Delta_{R}$ are the sample interval in angle and range dimensions, respectively. Actually, these two parameters can be neglected since they are constants once the number of discrete points are determined. Hence, Equation (25) can be reformulated as
(26)$$\min\sum_{p=1}^{P}\sum_{q=1}^{Q}W\left(R_{p},\theta_{q}\right)\left|P\left(R_{p},\theta_{q}\right)-P_{d}\left(R_{p},\theta_{q}\right)\right|\sin\left(\theta_{q}\right)$$

Finally, the fitness function can be constructed as
(27)$$f\left(b\right)=\sum_{p=1}^{P}\sum_{q=1}^{Q}W\left(R_{p},\theta_{q}\right)\left|P\left(R_{p},\theta_{q}\right)-P_{d}\left(R_{p},\theta_{q}\right)\right|\sin\left(\theta_{q}\right)$$

It should be highlighted here that the two elements located at the left-hand-side (LHS) and right-hand-side (RHS) boundaries of linear-FDA should be maintained to guarantee that the array aperture, which has a close relationship with the array resolution, is not changed after the sparse synthesis. Namely, $v_{m}$ in Equation (18) is equal to 1 when *n* = 1 and *n* = *N*, i.e., $v_{1}=1$ and $v_{N}=1$. In the following, the optimization procedures based on GA are provided to solve the above optimization problem in Algorithm 1, where *D* denotes the number of binary populations, *G* is the maximum generation, $N_{sparse}$ represents the element number of the sparse-FDA, $P_{c}$ and $P_{m}$ are the crossover and mutation probability, respectively. The complexity is $O\left(GN^{2}\left(N_{sparse}-2\right)\right)$. Note that the sparsity remains unchanged during the optimization procedures.
**Algorithm 1** Transmit–receive optimization procedures for Sparse-FDA.**Input:** $f_{0},d,D,N,N_{sparse},G,P_{c},P_{m}$**Output:** $b_{\mathrm{opti}}$Step 1. *D* transmit–receive weight vectors are initialized, which is coded with binary gene string under the constraint of $v_{1}=1$ and $v_{N}=1$. Let the generation g = 1.Step 2. Calculate the fitness value of each individual according to (27).Step 3. Select the optimal configuration parameters which can generate the minimum f(b).Step 4. Select descendent from the elites of the last generation.Step 5. Crossover and mutate the new generation according to the probability $P_{c}$ and $P_{m}$, respectively, while making sure that $N_{sparse}$ is a constant. Then update g = g + 1.Step 6. Repeat step 2 to step 5 until the maximum generation *G* is reached or a threshold is satisfied.Step 7. Determine the optimal transmit–receive weight vector $b_{\mathrm{opti}}$.

Once the optimal transmit–receive weight vector $b_{\mathrm{opti}}$ is obtained, the pattern of the sparse-FDA can be written as
(28)$$P_{sparse}\left(R,\theta \right)==b_{\mathrm{opti}}{}^{H}Q\left(R,\theta \right)b_{\mathrm{opti}},$$
consequently.

## 4. Numerical Results

In this section, simulation results are conducted to examine the properties of the proposed sparse-FDA. A uniform linear-FDA with N=M=60 antennas, i.e., full FDA, is considered before the sparse synthesis. Note that 10% array position error is considered in the simulations. The sparsity is 50%, i.e., 30 elements from the full FDA are selected to form the sparse-FDA. The whole observation region is set as Ω: 10 km≤R≤90 km, $0^{\circ }\leq \theta \leq 180^{\circ }$. The space of interest is located at $\left(R_{0},\theta_{0}\right)=\left(50\;\mathrm{km},90^{\circ }\right)$. Specific simulation parameters are provided in Table 1. The specific system parameters in GA are set as *G* = 100, *D* = 50, *P_c_* = 0.8 and *P_m_* = 0.05 according to [43,44]. Note that three kinds of frequency offsets presented in Figure 2 are considered. Figure 5a–c provides the optimal positions of the linear-FDAs with lin-FO, log-FO and sin-FO after the sparse synthesis, termed as lin-FO sparse-FDA, log-FO sparse-FDA and sin-FO sparse-FDA. As depicted in these figures, it is seen that a series of blue dots are utilized to indicate the selected elements while the yellow dots are adopted to indicate the unselected elements. Figure 5d presents the corresponding frequency offsets of the above three sparse-FDAs.

Figure 6 provides the 3D view and top views of the transmit–receive patterns of the full linear-FDAs with lin-FO, sin-FO and log-FO. As can be clearly seen in Figure 6a,d, three mainlobes are formed at the positions of (20 km, 90°), (50 km, 90°), (80 km, 90°) in the whole observation region. Namely, the full linear-FDA with lin-FO exhibits periodic transmit–receive pattern in angle dimension. Figure 6b,e show that the full linear-FDA with sin-FO has only one mainbeam in the whole range-angle space, which is the same case for the full linear-FDA with sin-FO. This demonstrate that the range-angle information decoupled pattern can be achieved for linear-FDA with nonlin-FO. Note that the mainlobe widths of full linear-FDAs with sin-FO and log-FO are 3.6 km and 7.2 km in range dimension, and 4° and 4° in angle dimension. Therefore, the mainlobe widths of these two FDAs during the sparse synthesis are set as 4.1 km and 5° for linear-FDA with sin-FO, and 7.7 km and 5.0° for linear-FDA with log-FO, which is slightly larger than their counterparts before the sparse synthesis.

Figure 7 shows the corresponding 3D view and top views of the transmit–receive patterns of the full linear-FDAs with lin-FO, sin-FO and log-FO after the sparse synthesis. As can be clearly seen in Figure 7a,d, three mainlobes are also emerged at the positions of (20 km, 90°), (50 km, 90°), (80 km, 90°) in the whole observation region. Namely, the full linear-FDA with lin-FO still exhibits periodic transmit–receive pattern in angle dimension after the sparse synthesis. Figure 7b,e show that the full linear-FDA with sin-FO also has only one mainbeam in the whole range-angle space after sparse synthesis, which is the same case for the linear-FDA with log-FO. This demonstrates that the range-angle information decoupled pattern can be maintained for linear-FDA with nonlin-FO after sparse synthesis. Compared Figure 7 with Figure 6, it is seen that the peak sidelobe levels (PSLLs) are slightly increased, and the mainlobe widths are slightly broadened after the sparse synthesis. In particular, the PSLLs of the full linear-FDAs with lin-FO, sin-FO and log-FO are $10\log_{10}\left(0.04281\right)=-13.7\mathrm{dB}$, $10\log_{10}\left(0.1073\right)=-9.7\mathrm{dB}$, and $10\log_{10}\left(0.04185\right)=-13.8\mathrm{dB}$. The PSLLs are $10\log_{10}\left(0.06138\right)=-12.1\mathrm{dB}$, $10\log_{10}\left(0.1097\right)=-9.6\mathrm{dB}$ and $10\log_{10}\left(0.05187\right)=-12.9\mathrm{dB}$ after sparse synthesis, which are only increase by 1.6 dB, 0.1 dB and 0.9 dB, respectively. The detailed PSLLs of these three linear-FDAs before and after the sparse synthesis are provided in Table 2. Table 3 provides the mainbeam width of the three FDAs in range and angle dimensions before and after the sparse synthesis. To summarize, the element number is essentially reduced, i.e., 50% of the elements are saved, only at the cost of a slightly PSLL increment.

To clearly compare the transmit–receive patterns of linear-FDA before and after the sparse synthesis, Figure 8 and Figure 9 depict the projection of the normalized patterns of linear-FDAs with log-FO and sin-FO in the separated range and angle domains. Comparing Figure 8a with Figure 8c, it is observed that the PSLL in angle dimension is increased form $10\log_{10}\left(0.03046\right)=-15.2\;\mathrm{dB}$ to $10\log_{10}\left(0.06191\right)=-12.1\;\mathrm{dB}$ for the linear-FDA with sin-FO after the sparse synthesis. Compared Figure 8b with Figure 8d, it is seen that the PSLL in angle dimension of the linear-FDA with log-FO is increased from $10\log_{10}\left(0.02315\right)=-16.4\;\mathrm{dB}$ to $10\log_{10}\left(0.03102\right)=-15.1\;\mathrm{dB}$ after the sparse synthesis. From Figure 9, it is seen that the PSLL in range dimension is increased by 0.1 dB (from $10\log_{10}\left(0.1073\right)=-9.7\;\mathrm{dB}$ to $10\log_{10}\left(0.1097\right)=-9.6\;\mathrm{dB}$) for the linear-FDA with sin-FO, and 0.9 dB (from $10\log_{10}\left(0.04185\right)=-13.8\;\mathrm{dB}$ to $10\log_{10}\left(0.05187\right)=-12.9\;\mathrm{dB}$) for the linear-FDA with log-FO, after the sparse synthesis. Table 4 provides the corresponding PSLLs of the pattern projection of the linear-FDAs with sin-FO and log-FO in range and angle dimensions. 

Figure 10 provides the fitness function values versus iterations for the sparse synthesis of linear-FDAs with nonlin-FO. It can be seen from this figure that the fitness function values decreased rapidly in the first 10 iterations and remained unchanged after 70 iterations.

Figure 11 presents the normalized transmit–receive patterns of linear-FDAs with sin-FO and log-FO profiles in range (cut at angle of 90°) and angle (cut at range of 50 km) dimensions before and after sparse synthesis. From Figure 11a,c, it is observed that the PSLL of the linear-FDA with sin-FO in angle dimension is increased by 15.8 dB (from −27.9 dB to −12.1 dB), while its counterpart in range dimension is increased by 0.1 dB (from −9.7 dB to −9.6 dB) after applying the sparse implementation. From Figure 11 b and Figure 9d, it is seen that the PSLL of the linear-FDA with log-FO in angle dimension is increased by 8.7 dB (from −26.2 dB to −17.5 dB), while its counterpart in range dimension is increased by 0.9 dB (from −13.8 dB to −12.9 dB) after applying the sparse implementation. Note that the performance of the two linear-FDAs with nonlin-FOs after sparse synthesis in the presence of 5% FO errors is investigated. It is seen that the PSLL is increased by within 0.3 dB, which also in turn demonstrates the effectiveness of the proposed approach. Table 5 provides the PSLLs of the profiles in Figure 11. In summary, the cost of linear-FDA can be essentially reduced at a slight cost of PSLL increment and broadened mainlobe width after sparse synthesis.

Table 6 provides the performance comparison of our method with other published works. Compared with the works presented in [39,41,42], it can be seen that the lowest PSLL can be achieved with the proposed approach under a large percentage of saved elements. Compared with the work presented in [35], the hardware complexity in this diagram is significantly reduced since only one mixer and one ADC are utilized.

## 5. Conclusions

To substantially reduce the cost while almost maintaining the antenna resolution, it is of importance to conduct an investigation on sparse synthesis for FDA. In this article, a transmit–receive sparse synthesis approach was presented for linear-FDA to achieve a time-invariant focused beampattern in a range-angle space. Specifically, a cost-effective transmit–receive signal processing diagram was adopted to eliminate the time-varying terms of linear-FDA with nonlin-FO. The corresponding formula was derived and analyzed in detail. Then, the sparse synthesis was implemented for the transmit–receive beamforming based on GA where the element positions were incorporated into the optimization problem. The numerical results show that 50% elements can be saved for the two linear-FDAs with nonlin-FOs, i.e., sin-FO linear-FDA and log-FO linear-FDA, with only a less than 1.0 dB increment in PSLL. In addition, the performance of the two sparse-FDAs in the presence of 5% FO errors was investigated. The results showed that the PSLL is increased by less than 0.3 dB, which in turn demonstrates the effectiveness of the proposed approach.

## Figures and Tables

**Figure 1 sensors-23-03107-f001:**
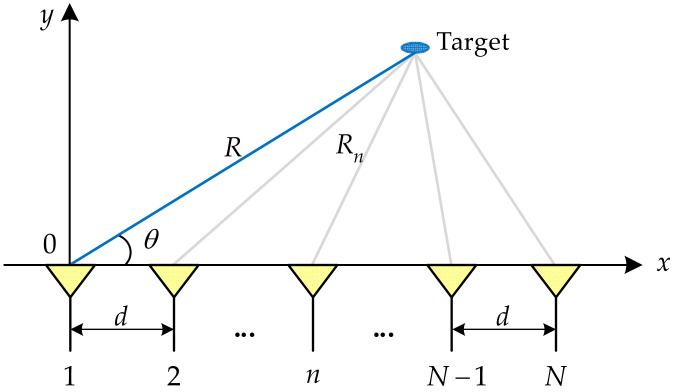
Configuration of a linear-FDA.

**Figure 2 sensors-23-03107-f002:**
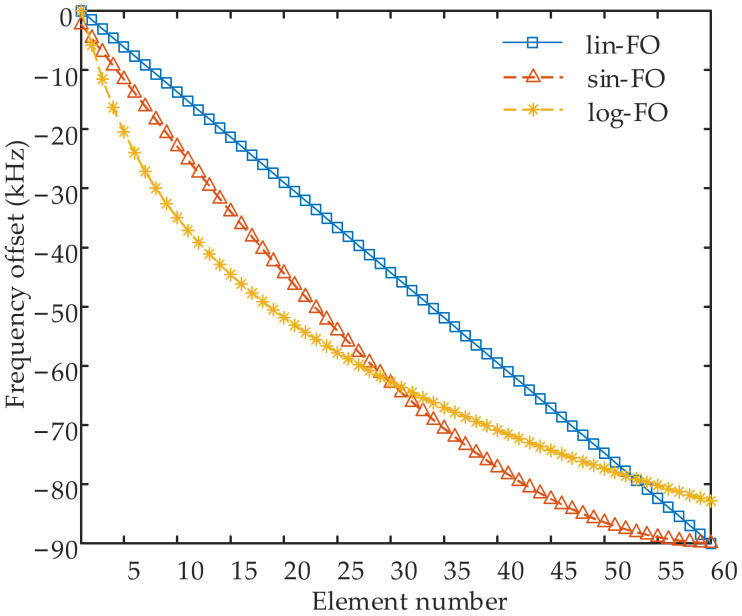
Comparative FOs with respect to element number, where *N* = 60 and Δf=10kHz.

**Figure 3 sensors-23-03107-f003:**
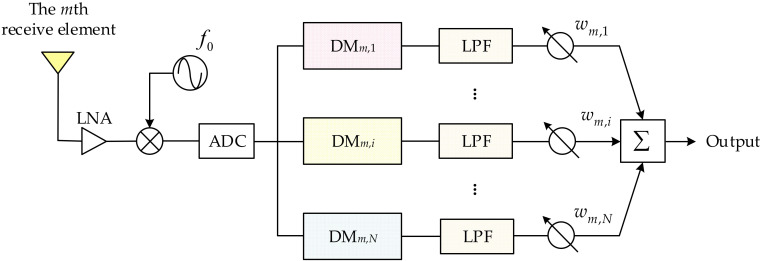
Diagram of receive signal processing chain of the *m*th receive element for linear-FDA with nonlin-OF (Figure 2b in [29]).

**Figure 4 sensors-23-03107-f004:**
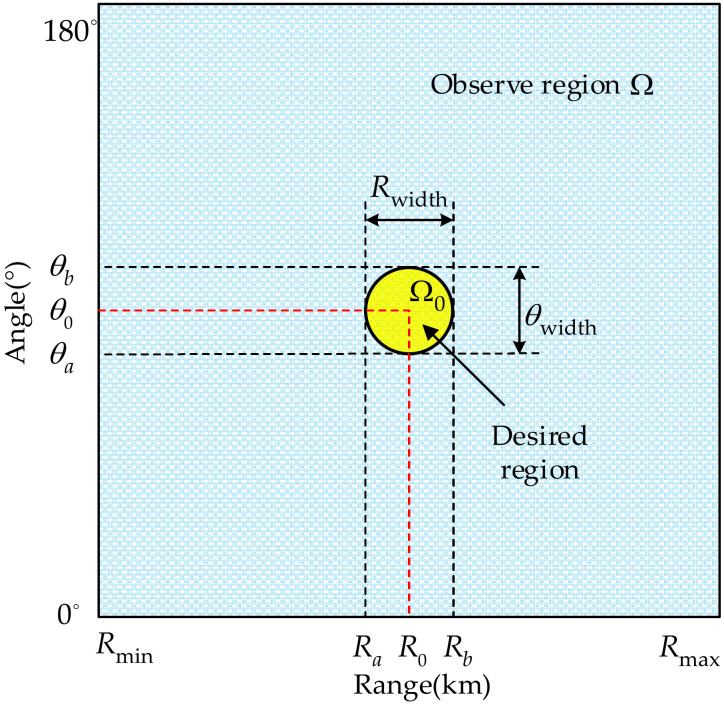
Illustration of area division of the whole observation in range-angle space.

**Figure 5 sensors-23-03107-f005:**
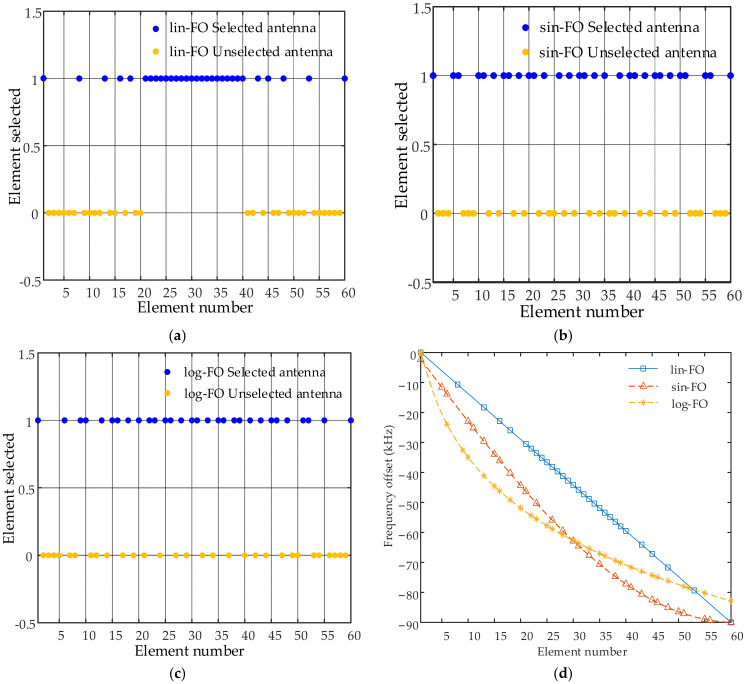
Optimal element positions and corresponding frequency offsets after sparse synthesis of linear-FDA: (**a**) optimal element positions of lin-FO sparse-FDA; (**b**) optimal element positions of sin-FO sparse-FDA; (**c**) optimal element positions of log-FO sparse-FDA; and (**d**) corresponding frequency offsets of the above three sparse-FDAs.

**Figure 6 sensors-23-03107-f006:**
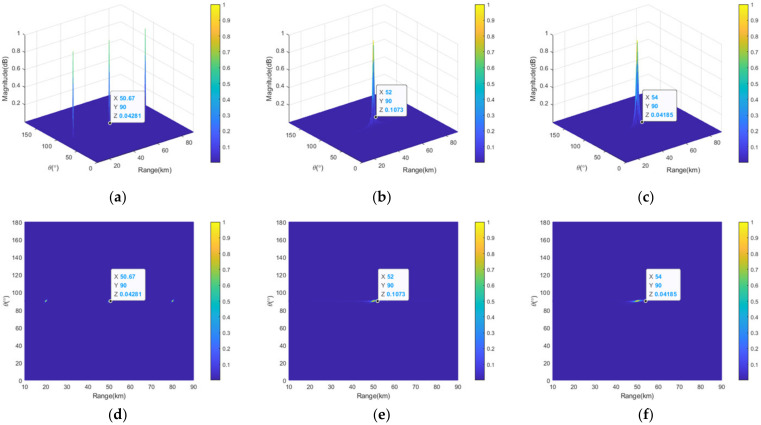
Comparative transmit–receive pattern of full linear-FDA with lin-FO: (**a**) 3D view; and (**d**) top view; with sin-FO; (**b**) 3D view; and (**e**) top view; and with log-FO (**c**) 3D view; and (**f**) top view.

**Figure 7 sensors-23-03107-f007:**
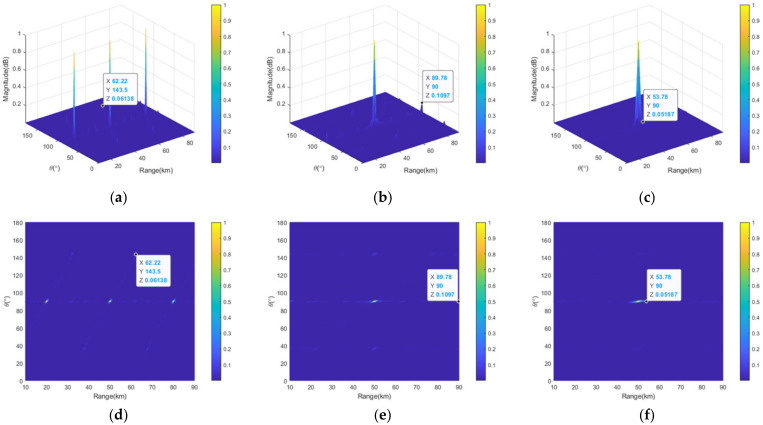
Comparative transmit–receive pattern of sparse-FDA with lin-FO: (**a**) 3D view; and (**d**) top view; with sin-FO; (**b**) 3D view; and (**e**) top view; and with log-FO; (**c**) 3D view; and (**f**) top view.

**Figure 8 sensors-23-03107-f008:**
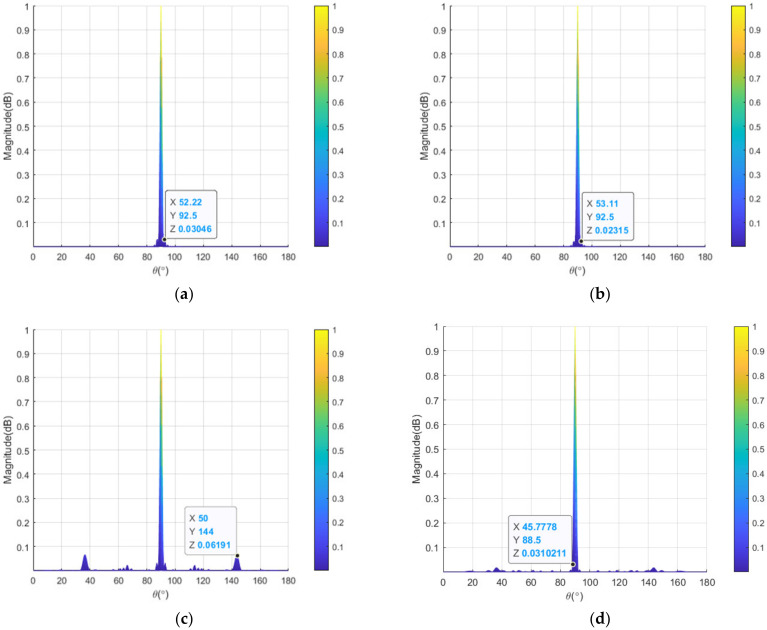
Projections of transmit–receive patterns in angle domain: (**a**) full linear-FDA with sin-FO; (**b**) full linear-FDA with log-FO; (**c**) sparse-FDA with sin-FO; and (**d**) sparse-FDA with log-FO.

**Figure 9 sensors-23-03107-f009:**
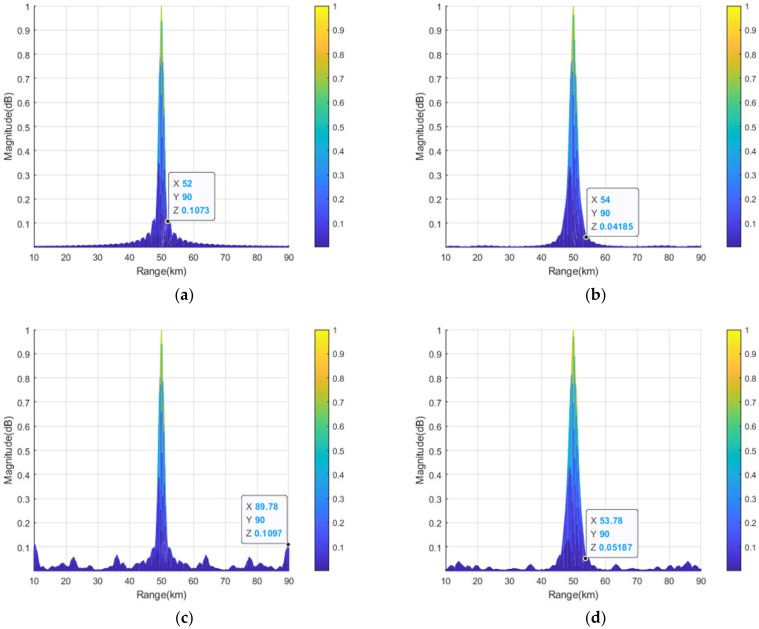
Projections of transmit–receive patterns in range domain: (**a**) full linear-FDA with sin-FO; (**b**) full linear-FDA with log-FO; (**c**) sparse-FDA with sin-FO; and (**d**) sparse-FDA with log-FO.

**Figure 10 sensors-23-03107-f010:**
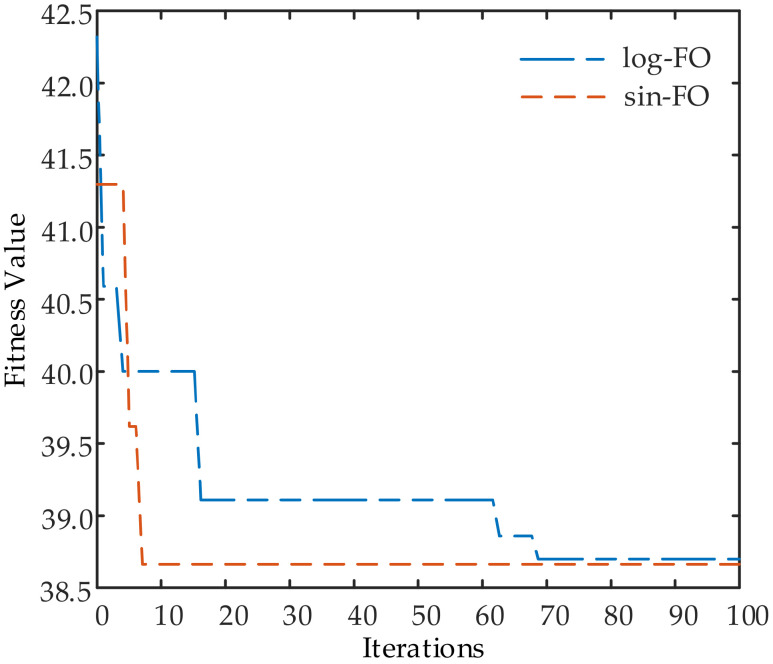
Fitness function value iterative curve of FDA sparse beamforming based on GA.

**Figure 11 sensors-23-03107-f011:**
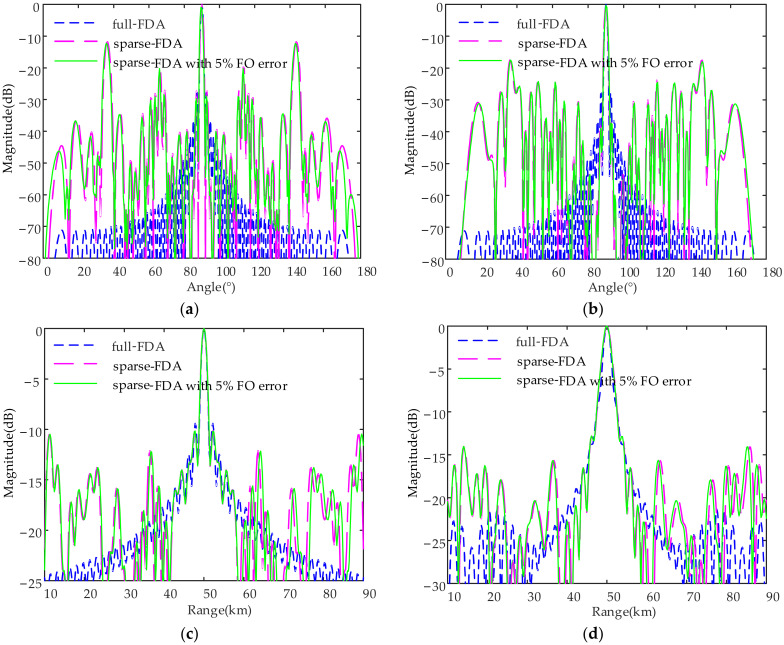
Normalized transmit and receive pattern profiles: (**a**) linear-FDA with sin-FO; and (**b**) linear-FDA with log-FO in angle domain at range of 50 km; (**c**) linear-FDA with sin-FO and (**d**) linear-FDA with log-FO in range domain at angle of 90°. Note that the profiles of linear-FDA after the sparse synthesis in the presence of 5% FO error are also provided as a comparison.

**Table 1 sensors-23-03107-t001:** Simulation parameters.

Parameters	Values	Parameters	Values	Parameters	Values
$f_{0}$	10 GHz	Δf	10 kHz	*N*	60
*d*	0.015 m	*G*	100	*D*	50
$P_{c}$	0.8	$P_{m}$	0.05	β	50%

**Table 2 sensors-23-03107-t002:** PSLLs comparison of transmit–receive pattern of the linear FDA before and after sparse synthesis.

	Full-FDA			Sparse-FDA		
	Lin-FO	Sin-FO	Log-FO	Lin-FO	Sin-FO	Log-FO
PSLL (dB)	−13.7	−9.7	−13.8	−12.1	−9.6	−12.9

**Table 3 sensors-23-03107-t003:** Mainbeam width comparison of transmit–receive pattern of the linear FDA before and after sparse synthesis.

	Full-FDA			Sparse-FDA		
	Lin-FO	Sin-FO	Log-FO	Lin-FO	Sin-FO	Log-FO
Angle (°)	4.0	4.0	4.0	10.0	5.2	5.0
Range (km)	1.3	3.6	7.2	3.6	4.4	8.0

**Table 4 sensors-23-03107-t004:** PSLLs comparison of pattern projection of linear FDA in range and angle dimensions before and after sparse synthesis.

	Full-FDA		Sparse-FDA	
	Sin-FO	Log-FO	Sin-FO	Log-FO
Angle (dB)	−15.2	−16.4	−12.1	−15.1
Range (dB)	−9.7	−13.8	−9.6	−12.9

**Table 5 sensors-23-03107-t005:** PSLLs comparison of pattern profiles in Figure 11.

	Linear-FDA with Sin-FO			Linear-FDA with Log-FO		
	Full-FDA	Sparse-FDA	With 5% FO Error	Full-FDA	Sparse-FDA	With 5% FO Error
Rang (dB)	−9.7	−9.6	−9.5	−13.8	−12.9	−12.8
Angle (dB)	−27.9	−12.1	−12.0	−26.2	−17.5	−17.3

**Table 6 sensors-23-03107-t006:** Performance comparison with other published works.

	Mode	Scheme Complexity	Percentage of Saved Elements	PSLL of Full-FDA (dB)	PSLL of Sparse-FDA (dB)
[39]	T	/	40%	−6.6	−7.1
[41]	T	/	70%	−5.5	−5.5
[42]	T	/	20%	−7.6	−7.5
[35]	T-R	Complicated	/	/	/
Proposed	T-R	Simple	50%	−9.7/−13.8	−9.5/−12.8

Note that “T” denotes the “Transmit mode” while “T-R” stands for “Transmit–receive mode”.

## Data Availability

Not applicable.

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
