# Peer review of "Transmit–Receive Sparse Synthesis of Linear Frequency Diverse Array in Range-Angle Space Using Genetic Algorithm"

_sensors, 2023, doi:10.3390/s23063107_

Round 1
Reviewer 1 Report
This paper investigates transmit-receive sparse synthesis approach for linear-FDA. The number of antenna elements can be reduced by half by using this method, while maintaining good spatial focusing performance. Even through the PSLL has increased slightly after optimization, the proposed method has its advantage because it is more cost-effective.
Author Response
Thanks for your positive evaluation of our work. To improve its presentation and readability, the manuscript has been carefully checked through to fix some grammatical errors.

Reviewer 2 Report
This paper proposes a cost-effective transmit-receive sparse synthesis approach of linear-FDA in range-angle space. With the proposed approach, the number of array element can be greatly reduced while maintaining the time-invariant pattern with good spatial focusing performance. I found your ideas very useful for radar implementations. I have only few minor comments.
1. It is may be better to add the convergence property of the proposed approach in Section 4.
2. The specific constraints on the mainbeam width in range and angle dimensions during the optimization should be provided in Section 4.
3. Please consider to gather in a table for the PSLLs results of Figures 7-9. It will be easier to follow the results presentation.
4. There are several grammatical errors in the manuscript. The authors should check the manuscript through and fix such errors.
Author Response
Thanks for your positive evaluation of our work. Your comments are valuable for us. Our corresponding responses to your comments are provided in the following word file.

Reviewer 3 Report
The authors should try to write one strong novelty in the paper instead of writing three points.

Author Response
We appreciate your time and efforts in evaluating our manuscript. Your comments are valuable for us. Our responses to your comments are provided in the following word file.

Reviewer 4 Report
This paper presents proposes a transmit-receive sparse synthesis approach of linear-FDA in range-angle space. By using the proposed method, the number of antenna elements can be reduced by half while maintaining the time-invariant range-angle information decoupled transmit receive pattern with good spatial focusing performance.
My comments can be summarised as below:
1. In the "Abstract", authors should provide the performance quantitative achievement of the proposed method.
2. In the "Introduction", authors claimed that the contributions of the proposed work can be highlighted in three areas as below.
(i) A new cost-effective transmit-receive signal processing scheme is established to handle the inherent time-varying property of FDA with low hardware complexity since only a mixer, an analog-to-digital converter (ADC) and a series of identical low-pass filters (LPFs) are utilized.
(ii) Based on the above established transmit-receive scheme, two linear-FDAs with non- linearly varying frequency offset (nonlin-FO) are proposed to achieve range-angle information coupled transmit-receive pattern in range-angle space.
(iii). The theoretical model is constructed to realize transmit-receive sparse synthesis of 128 linear-FDA based on GA with focused mainbeam in desired range-angle space and low SLL in the region out-of-interest simultaneously
But, it is not clear how well the proposed method compared to other published works in the literatures. Authors should compare their works with others. Perhaps a table to show this will be useful to justify their contribution claims.
3. In section "Numerical Results", how authors selected properties of the proposed sparse-FDA and others ?? In Table 1, why specific simulation parameters were selected ?? Can these be other numbers ??
4.Different circumstance parameters should be introduced into the experiments/analysis. The reliability, stability and repeatability of the method and experiments should be analysed and discussed.
5. Any error analysis in the experiment in section 4: Numerical results ?? This should be discussed in the obtained results.
6. How authors verify the presented numerical results ?? How the presented results agreed or disagreed with the published results. Any experimental works to validate the presented results ?
7. Why GA optimisation method was selected for this analysis ?? Why not other optimisation method ?? Can other method achieves a better results ?
8. The conclusion of this paper is very poor. Authors should rewrite and improve it by including qualitative achievement of proposed method in this section.
Author Response
We appreciate it your time and efforts in evaluating our manuscript. Your comments are valuable for us. We have carefully revised our manuscript according to your comments. The following provides our detailed responses.

Round 2
Reviewer 4 Report
Authors have addressed all my comments. I have no further comment.